# Usefulness of Probing Sensor Device for Evaluating Meniscal Suture and Scaffold Implantation

**DOI:** 10.3390/biomimetics9040246

**Published:** 2024-04-19

**Authors:** Shunsuke Sezaki, Shuhei Otsuki, Takashi Ishitani, Takeru Iwata, Takehito Hananouchi, Yoshinori Okamoto, Hitoshi Wakama, Masashi Neo

**Affiliations:** 1Department of Orthopedic Surgery, Osaka Medical and Pharmaceutical University, 2-7 Daigakumachi, Takatsuki 569-8686, Japan; shunsuke.sezaki@ompu.ac.jp (S.S.); takashi.ishitani@ompu.ac.jp (T.I.); takeru.iwata@ompu.ac.jp (T.I.); yoshinori.okamoto@ompu.ac.jp (Y.O.); hitoshi.wakama@ompu.ac.jp (H.W.); neo@ompu.ac.jp (M.N.); 2QOL Research Division, GUNZE MEDICAL Ltd., Kita-ku, Osaka 530-0003, Japan; 3Biodesign Division, Department of Academia-Government-Industry Collaboration, Hiroshima University, Minami-ku, Hiroshima 734-8551, Japan; takehito@hiroshima-u.ac.jp

**Keywords:** probing sensor, meniscus, scaffold, meniscal repair, vertical tear

## Abstract

Appropriate suture tension is a key factor in successful meniscal repair. This study aimed to clarify the appropriate value of meniscal stabilization with suture repair based on a probing procedure for healthy porcine menisci and a novel meniscal scaffold. After evaluating the reliability of the probing sensor, meniscal vertical tear and partial meniscectomy models were developed, in which suture repair and meniscal scaffold implantation were performed at suture intervals ranging between 20 and 2.5 mm. The residence forces at each interval were evaluated using a probing sensor. Moreover, a tensile test was conducted to evaluate the displacement and presence or absence of gaps. We found that normal and meniscal scaffolds should be fixed within 5 mm of suture interval. The probing residence forces required were at least 1.0 N for vertical tears and 3.0 N for meniscal scaffolds. These findings may be taken into consideration to reduce suture failure following meniscal tear repair and stabilizing meniscal scaffold fixation.

## 1. Introduction

The menisci of the knee joint are crescent-shaped wedges of fibrocartilage that play critical roles in shock absorption, load transmission, joint stability, lubrication, and proprioception at the joint [1,2,3,4]. Damage to the meniscus is one of the most common injuries to the knee joint, often resulting from an impact during sports participation or simply joint degeneration [5]. The strategy for “Save the meniscus” has been generally recognized [6,7] because meniscal loss could be regarded as a pre-arthritic condition for the knee due to loss of protective function [8]. Meniscal injury is mainly treated with suture repair to preserve the meniscal size and function. On the other hand, irreparable meniscal tear treatment performed using partial meniscectomy, which significantly increases peak contact pressure across the tibiofemoral joint compartment, reportedly leads to degenerative changes and osteoarthritis [9,10]. To resolve this issue, meniscal scaffolds have been developed [11,12,13,14] and are indicated for irreparable meniscal tears to preserve the meniscal size and function [15,16].

Appropriate suture tension is important for avoiding suture loosening or cheese wiring after meniscal repair [17]. This is because 15–30% of patients experience re-tearing after meniscal suture repair [18]. Additionally, 10% of patients require re-operation after meniscal scaffold implantation [19,20]. Although regulating suture tension for meniscal repair or implantation is essential to improve the success rate of meniscal sutures, there is no device that can quantitatively assess the suture tension in clinical practice. Therefore, suture tension for meniscal repair or implantation only relies on “surgeon’s sense,” and controlling suture tension remains a challenge.

Probing is performed during arthroscopic surgeries to identify the injured lesion, instability with gap appearance in the repair area, and repair tension of the soft tissue in the knee [21,22]. Recently, a probing device was developed to assess the hip and knee joints [23,24] quantitatively. This probing device quantitatively measures the resistance of soft tissues with a triaxial force sensor; it has been reported to be useful for measuring some mechanical properties of the joints during arthroscopy [23]. Therefore, this device may potentially determine the appropriate suture tension for meniscal repair.

In the current study, we focused on the tension of meniscal suture repair and scaffold fixation force using a probing sensor. We hypothesized that meniscal suture repair and scaffold implantation using a controlling probing sensor might accomplish appropriate suture repair tension and reduce gap instability. Therefore, as a basic study, this study aimed to evaluate whether the probing sensor could be a reliable device for assessing its clinical application and usefulness. We also aimed to determine the appropriate pull-out strength of this device during meniscal suture repair and scaffold implantation using porcine meniscus.

## 2. Materials and Methods

### 2.1. Probing Sensor

Probing force was evaluated using a probing device equipped with a triaxial force sensor (Probing Sensor, Takumi Precise Metal Work Manufacturing Ltd., Osaka, Japan) (Figure 1a). The probing sensor comprised a probe measuring 150 mm in length with a tip of 3 mm, a handle, and an embedded sensor. The probing force could be calculated in all three directions (*x*, *y*, *z*) and was defined as the value calculated using the following formula, considering that the axes may shift slightly for each test in the current study:(1)$$F=\sqrt{x^{2}+y^{2}+z^{2}}$$
where F is the probing force, and *x*, *y*, and *z* are the forces in each direction (Figure 1b).

### 2.2. Reliability of the Probing Sensor

A universal testing machine (Autograph AGS-X, Shimadzu Corporation, Kyoto, Japan) with a load cell of 100 N was used to evaluate the probing sensor’s reliability. The universal testing machine was used to assess materials’ tensile and compressive strengths. The load cells used in this test were calibrated annually. Therefore, the values calculated by the universal testing machine were accurate. First, the handle of the probing sensor was fixed using a jig installed at the bottom of the universal testing machine; the probing sensor was set vertically. A loop was then made using a 2-0 Ethibond suture (ETHICON Inc., Bridgewater, NJ, USA). One end of the suture was hooked onto the tip of the probe, and the other end was attached to the jig of a universal testing machine to form a loop. Loads equivalent to 3.0, 5.0, and 10.0 N were applied to the probing sensor at a speed of 1.0 N/s by the universal testing machine via the 2-0 Ethibond suture (Figure 2). The tests were performed at least five times for each load. The tensile force the universal testing machine applied and the mean loads calculated from the device were compared. The intraclass correlation coefficient (ICC) of intraobserver reliability was computed using the load calculated from the device to demonstrate the reliability of the evaluation.

### 2.3. Evaluation of Appropriate Suture Interval for Meniscal Tear

#### 2.3.1. Evaluation of Tensile Strength Using Probing Sensor

Fresh five porcine lateral menisci, aged 6 months, were used in this study. The porcine knee used for the experiment was purchased from the slaughterhouse (Kyoto Kyodo Kanri Co., Ltd., Kyoto, Japan) and was originally intended for disposal. In this circumstance of purchase, approval of the Ethical Committee was not required, hence the absence of information on the matter. The femur was removed to expose the meniscus, and a 25 mm vertical tear was created in the meniscus using scalpel #11 (Figure 3a). The 25 mm width tear was then sutured 5 mm from the end using 2-0 FiberWire (Arthrex Ltd., Naples, FL, USA), making it a 20 mm width tear. After suturing, the tip of the probe was hooked onto the 20 mm wide tear, and the load was calculated when the probe was pulled 5 mm quantitatively in the *z*-direction of the probing sensor (Figure 3b). After obtaining data using the probing sensor, suture fixation was performed at a position 5 mm from the end of the 20 mm width tear to create a 15 mm width tear, and data were calculated using the same method using the probing sensor. Similarly, tears with widths of 10, 5, and 2.5 mm were created, and loads measured by the probing sensor were calculated. The probing force was calculated from the average of three quantitative tensile tests for each test.

#### 2.3.2. Evaluation of Meniscal Suture Strength Using Biomechanical Test

To investigate the relationship between the width of tear and displacement of meniscus during loading, a biomechanical test was performed using a universal testing machine with a cell load of 100 N to assess displacement during constant loading. Using the porcine meniscus sample with a 2.5 mm tear width, retention sutures (2-0 FiberWire) were hooked at two locations on the meniscal rim and one at the center of the inner edge; a loop was formed. The looped suture was attached to the pin of the universal testing machine jig to form a loop, and the outer and inner rim sutures were placed on the upper and lower jigs of the universal testing machine, respectively. The biomechanical test was conducted under conditions where a preload of 0.1 N was applied, and the load was applied at a speed of 30 mm/min until 10 N was reached (Figure 4). Displacement was calculated after reaching 10 N.

After testing with a 2.5 mm tear width, samples with 5 mm tear width were prepared by cutting the suture. Subsequently, a biomechanical test was performed in the same manner as mentioned above, and displacement at 5 mm fracture width was calculated. The aforementioned process calculated displacements for tear widths of 10, 15, and 20 mm (*n* = 5).

For each test, the meniscal samples were photographed using a digital camera (Nikon COOLPIX P300; Nicon Corporation, Tokyo, Japan) before the test and after reaching 10 N to evaluate the appearance of a gap. If a gap was observed in the crack, it was defined as the gap group; otherwise, it was defined as the non-gap group.

### 2.4. Evaluation of Appropriate Suture Interval for Meniscal Scaffold Implantation

#### 2.4.1. Fabrication of the Meniscal Scaffold

The meniscal scaffold comprised polyglycolic acid (PGA) and L-lactide-ε-caprolactone copolymer (P(LA/CL)). This novel scaffold was manufactured using PGA felt (NEOVEIL^®^; Gunze Medical Ltd., Osaka, Japan) coated with a P(LA/CL) sponge fixed using PGA sutures (Gunze Medical Ltd., Osaka, Japan). PGA laminate was immersed in a solution of copolymer (polyester with a 50:50 molar composition of lactate and ɛ-caprolactone) in 1,4-dioxane and freeze-dried (DRZ350WA; ADVANTEC Co, Ltd., Tokyo, Japan). The inner PGA laminate and outer sponge layers were approximately 6 to 6.5 mm and 100 to 400 µm thick, respectively. The scaffold was trimmed to form the final C-shaped scaffold, with the average size of the meniscus in the porcine specimen: diameter, 40 mm; body width, 15 mm (Figure 5) [25,26].

#### 2.4.2. Evaluation of Suture Strength in Meniscal Scaffold Implantation Using Probing Sensor

A meniscal defect was created using #11 scalpel to simulate a typical meniscectomy in the central zone. The size of the meniscal defect was uniform at 80% body width and a width of 20 mm (Figure 6a).

The meniscal scaffold was trimmed to the same size as the defect and implanted into it (Figure 6b). After the implanted scaffold was fixed in two places at each end using 2-0 FiberWire (Figure 6c), a probing sensor was hooked between the scaffold and the meniscal rim. Thereafter, a fixed amount of tension (5 mm) was applied to the center of the knee. Subsequently, suture fixation was performed at a position 5 mm from the edge such that the unsutured part of the scaffold and meniscal rim were 15 mm wide, and the load at 5 mm quantitative tension was determined using a probing sensor. Using the same process as described above, the unsutured parts of the scaffold and meniscal rim were sutured to widths of 10, 5, and 2.5 mm, and the quantitative load at each interval was calculated using a probing sensor (*n* = 5). The probing force was calculated from the average of three quantitative tensile tests for each test. 

#### 2.4.3. Evaluation of Suture Fixation of Meniscal Scaffold Using Biomechanical Test

Biomechanical testing was performed using a universal testing machine to investigate the relationship between the width of the unsutured meniscal scaffold and the presence or absence of a gap. Using the samples evaluated in Section 2.4.2, installation on a universal testing machine, tensile testing, and calculation of test data were performed using the same process described in Section 2.3.2. After testing with a suture fixation width of 2.5 mm, displacements were also calculated for suture fixation widths of 5, 10, 15, and 20 mm, as described in Section 2.3.2 (*n* = 5). For each test, images were captured before and after the test using a digital camera, and the gap and non-gap groups were defined based on the presence or absence of a gap in the unsutured area.

### 2.5. Statistical Analysis

All data analyses were performed using JMP Pro, version 15 (SAS Institute Japan Ltd., Tokyo, Japan). The calculated data are expressed as average ± standard deviation. Dunn’s multiple comparison method was used to analyze the differences between groups. Statistical significance was set at *p* < 0.05.

## 3. Results

### 3.1. Reliability of the Probing Sensor

A universal testing machine applied loads equivalent to 3.0, 5.0, and 10.0 N to the probing sensor using the 2-0 Ethibond suture, resulting in a calculated load of 2.99 ± 0.16, 4.66 ± 0.13, and 9.64 ± 0.17 N from the probing sensor, respectively (Figure 7). The loads applied by the universal testing machine and those calculated by the probing sensor correlated with each other, with an ICC of 0.99.

### 3.2. Evaluation of Suture Strength Using Probing Sensor in Porcine Menisci

The macroscopic findings did not reveal any gap in the meniscal sutures with less than 5 mm suture interval (Figure 8a). The probing sensor values were less than 1.0 N without a significant difference between 10 to 20 mm suture interval. The probing force was 1.4 ± 0.5 N at a suture interval of 5 mm and 3.7 ± 0.9 N at 2.5 mm, which was significantly higher compared with the suture interval between 10–29 mm (*p* < 0.05, Figure 8b). The amount of displacement by tensile testing was not significantly different from that of the normal porcine meniscus up to a suture spacing of 10 mm. In addition, the amount of displacement was significantly greater than that of the porcine meniscus at suture spacings of ≥15 mm. (*p* < 0.05, Figure 8c). The relationship between the results of a suture force test using a probing sensor and displacement results after the tensile test using the universal testing machine was inversely proportional (Figure 8d).

### 3.3. Evaluation of Suture Strength Using Probing Sensor in Meniscal Scaffold

Regarding the macroscopic findings described in Section 3.2, no gaps were detected in the meniscal sutures with a suture interval of less than 5 mm (Figure 9a). The probing sensor values were 1.9 ± 1.2 N at 10 mm suture interval, 3.7 ± 1.7 N at 5 mm suture interval, and 4.8 ± 1.3 N at 2.5 mm suture interval. Significant differences were detected between 2.5 mm and >10 mm and between 5 mm and 15 mm (*p* < 0.05, Figure 9b). Displacement measured using tensile testing differed significantly from the normal porcine meniscus at all suture intervals. Although there was a trend towards greater displacement with increasing suture interval, no significant difference was observed between 2.5 mm and 15 mm (*p* > 0.05). Significant differences were detected between 2.5 mm and 20 mm (*p* < 0.05, Figure 9c). Considering the scaffold fixation, displacement values were 8.2 ± 2.0 mm at 2.5 mm suture interval, 8.6 ± 2.0 mm at 5 mm suture interval, 9.4 ± 1.4 mm at 10 mm suture interval, 10.0 ± 1.2 mm at 15 mm suture interval, and 12.2 ± 2.1 N at 20 mm suture interval. Similar to the finding presented in Figure 8d, the relationship between the results of a suture force test using a probing sensor and dis-placement results after a tensile test using the universal testing machine was inversely proportional (Figure 9d).

## 4. Discussion

We found that the probing sensor was reliable for evaluating meniscal sutures. Moreover, to the best of our knowledge, this study is the first to identify the residual force of appropriate suture tension in normal meniscus and meniscal scaffold using a probing sensor. Our findings also indicated that this method may be useful for application in clinical practice in the future.

First, the validity of using the resultant force of three axes as the calculated value was evaluated. This sensor could calculate the force in all three axial directions. However, a slight deviation in the loading direction might affect the load in each direction based on clinical use. Therefore, the resultant force of the three axes was defined as the calculated value, and its correlation with the applied load was evaluated. As a result, the calculated resultant force was highly reliable and considered a useful evaluation method. 

Meniscal sutures are commonly used in clinical practice. However, studies determining the interval between sutures and tension that sutures could be sustained are scarce. Meniscal suture tension is important because the meniscus moves from an anterior to posterior position during knee motion, and this movement is more severe in weight-bearing situations [27]. Although several suturing techniques and devices have been developed [28], the re-operation rate remains more than 20% after meniscal suturing [17,19], indicating that appropriate meniscal suture technique might be critical to reducing the re-operation rate. In the current study, tensile stress was adapted based on the pull-out strength under 10 N because Dürselen et al. had reported that the meniscus could be sutured sufficiently with a pull-out strength device of 10 N or less [29]. In addition, the suturing interval for vertical tears should be within 5 mm based on gap observation with tensile stress in normal menisci. This is consistent with a previous study that reported a 3–4 mm interval for vertical repair [30]. 

To date, analysis of suture fixation tension was performed qualitatively using arthroscopy, termed “surgeon’s feeling.” The resistance force of the meniscal vertical tear after repair was measured, and it was clearly shown that safety fixation without a gap increased the residual force when tensile stress was applied. Of note, the sensor value reached more than 1 N when the suture interval was within 5 mm of the suture interval. This information is critical for identifying appropriate suture intervals. It was thought that this device could turn qualitative arthroscopy, termed “surgeon’s sense”, into a quantitative evaluation. Based on this data, if clinically appropriate pull-out strength is evaluated in further research using this device, the re-operation rate after meniscus suturing may be reduced. Care should be taken in the context that sutures should not be too tight to avoid the occurrence of cheese-wiring.

Meniscal scaffolds have been developed and studied to treat irreparable meniscal tears and degeneration [31]. The failure rate of meniscal scaffolds has been reported to range between 0 and 18% [32,33]; this statistic also includes suture fixation failure. Recently, we established a meniscal scaffold comprising PGA coated with polylactic acid/caprolactone [P(LA/CL)] [25]. A clinical trial was completed based on the aforementioned findings [34]. However, one patient demonstrated cheese wiring during suture fixation, suggesting that the appropriate implant size and suture tension required further investigation. The current study also analyzed the tension of scaffold fixation, which was similar to normal meniscus fixation. Normal menisci and meniscal scaffold experiments indicated that a suture interval of less than 5 mm was acceptable based on the probing sensor’s gap appearance and value. This information indicated that, similar to meniscal repair, the use of this device in clinical practice may provide superior fixation for meniscal scaffold grafts and reduce their failure rate.

In the context of probing forces between the gap and non-gap groups in vertical tear, size > 10 mm was elongated, and the gap increased by approximately 150% compared with non-gap in normal menisci. On the contrary, scaffold implantation resulted in approximately 120% elongation, which was less than normal meniscus fixation, indicating that the scaffold was stiffer and could result in cheese wiring as a complication rather than native menisci. The probing device might be helpful in determining the suture interval and simultaneously in controlling suture tension. 

In other areas of orthopedics, as mentioned in the background, this device is being applied to quantitatively evaluate the hip [23] and knee joints [24]. Regarding applied use to the hip joint, no difference in the stiffness of the hip labrum between in vivo and cadaveric tissue has been reported; the calculated value using the probing device was 4.2–8.3 N [23]. On the other hand, in a study evaluating anterior cruciate ligament (ACL) reconstruction, a moderate correlation between the amount of strain caused by the resistance force of the ACL using a probing sensor and the tensile tension of the ligament was shown; the value calculated using the probing device was 0.67–1.54 N [24]. Although a simple comparison is impossible, the appropriate pull-out strength for meniscal repair or scaffold fixation obtained in our study may be a reasonable value, considering that the meniscus was a cartilage-like tissue. Additionally, Tuijthof et al. reported in a probing force test using a meniscal model that forces exceeding 8.5 N should be avoided when probing meniscal tissue [35]. There may also be room to consider the upper limit of probing force.

The present study has a few limitations that need consideration. Normal menisci of young pigs were used in the experiment, which differs from human menisci. Therefore, an experimental system resembling the human knee environment should be established, as biomechanical results may differ from physiological loads in humans. The experimental tear model designed using porcine menisci was not a physiological injury because several meniscal tears were clinically observed. Generally, arthroscopic surgery was performed for meniscal treatment in different situations, which may have affected the probing forces. Irrespective of the limitations, appropriate suture tension for meniscal tears could have the potential to reduce the re-operation rate, which can be regulated with a probing sensor.

## 5. Conclusions

Normal and meniscal scaffolds should be fixed within 5 mm of suture interval, and the probing residence forces required were at least 1.0 N for vertical tears and 3.0 N for meniscal scaffold. This information demonstrated adequate meniscal suture fixation force through in vitro evaluation. With further investigation, this device may be able to contribute to reducing the failure rate of meniscal suture repair and meniscal scaffolding in clinical practice.

## Figures and Tables

**Figure 1 biomimetics-09-00246-f001:**
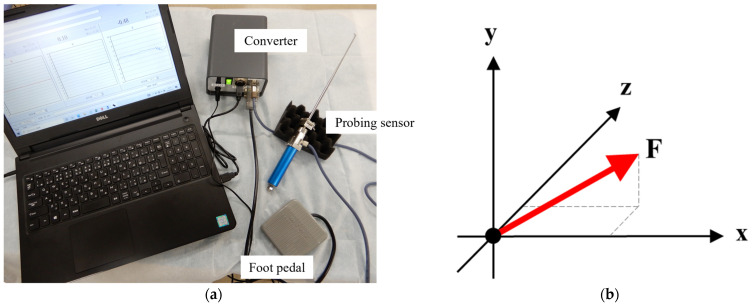
(**a**) Probing sensor system. (**b**) Representative diagram of probing force calculation. The red arrow indicates probing force.

**Figure 2 biomimetics-09-00246-f002:**
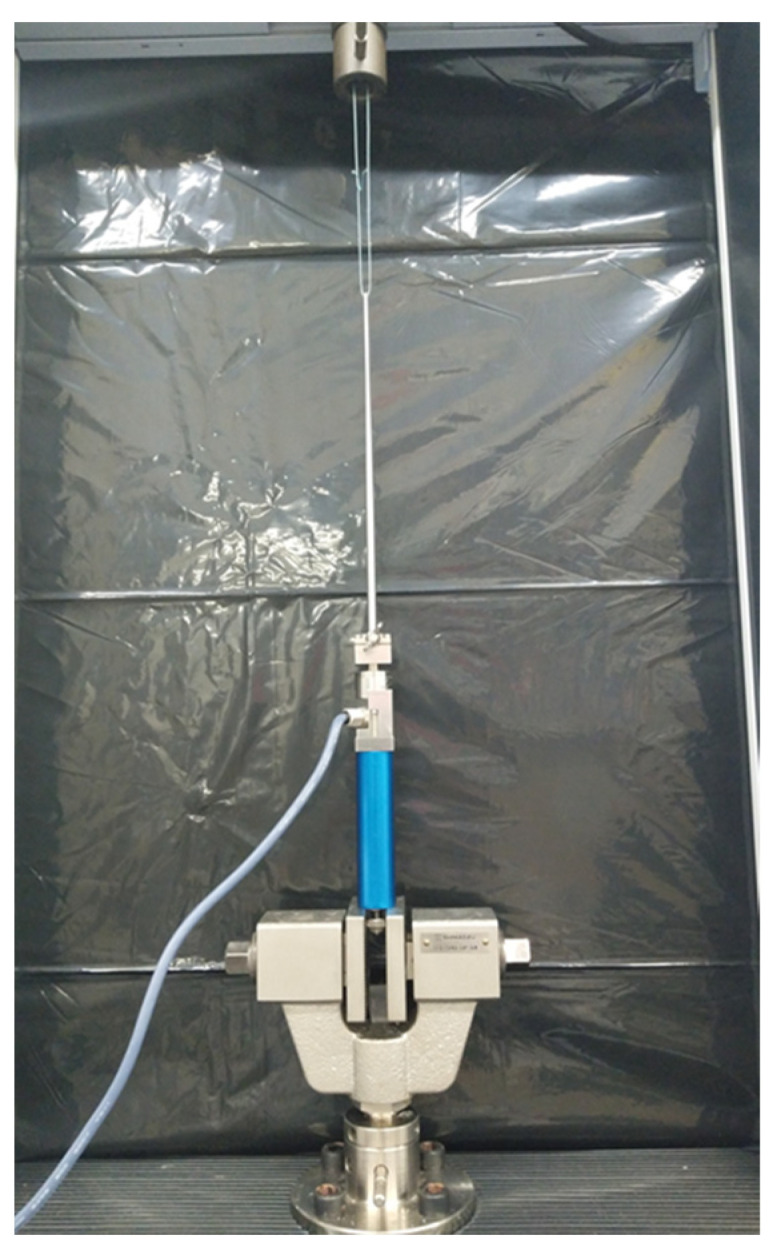
Reliability of the probing sensor. Tensile test performed using a 2-0 Ethibond suture in the universal testing machine.

**Figure 3 biomimetics-09-00246-f003:**
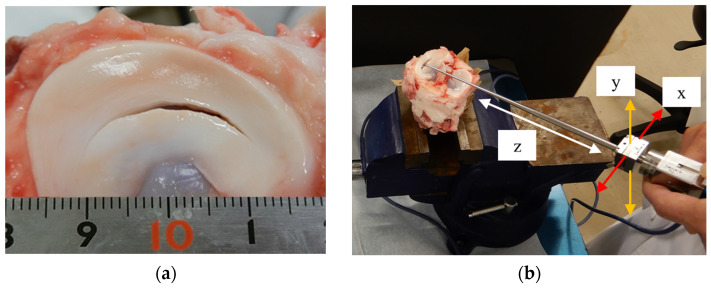
(**a**) Image of a porcine lateral meniscus with a 25 mm vertical tear created using scalpel #11. (**b**) Overall representation of tensile test in which 5 mm quantitative tension is applied in the *z*-direction.

**Figure 4 biomimetics-09-00246-f004:**
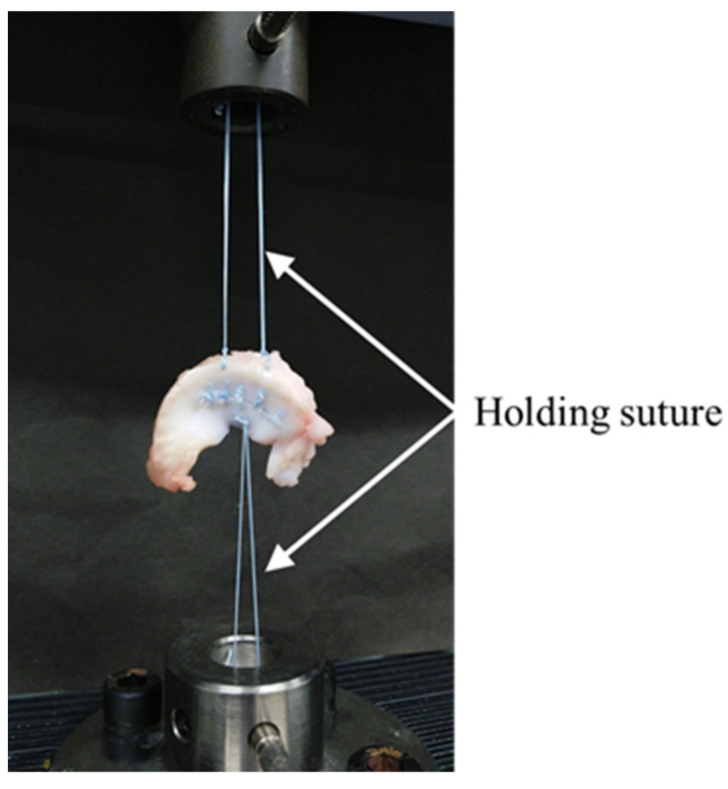
Biomechanical test being performed using the universal testing machine after suturing a porcine meniscus. The white arrow indicates 2-0 FiberWire for hold.

**Figure 5 biomimetics-09-00246-f005:**
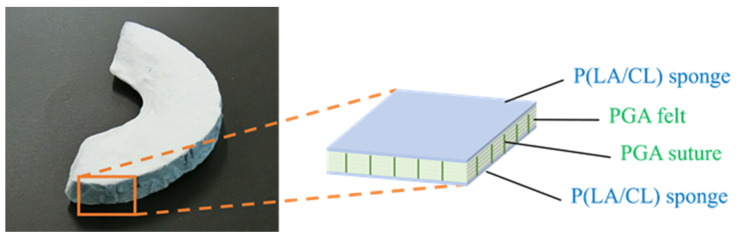
A meniscal scaffold comprising polyglycolic acid (PGA) covered with L-lactide-ε-caprolactone copolymer (P(LA/CL)). Overall image and configuration diagram of the scaffold. The device is cut according to the average porcine medial meniscus size.

**Figure 6 biomimetics-09-00246-f006:**
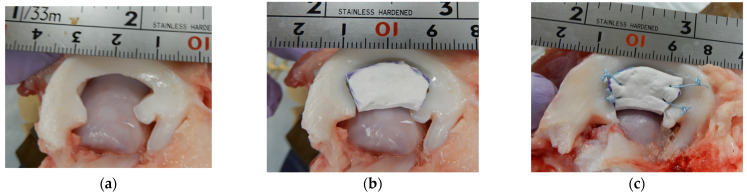
(**a**) Porcine lateral meniscus with a 20 mm defect was created. (**b**) The meniscal scaffold is trimmed to fit the defect size. (**c**) The meniscal scaffold is secured using two sutures at each end. A similar examination, as shown in Figure 3b, is performed to evaluate the relationship between the unstitched margin and probing force.

**Figure 7 biomimetics-09-00246-f007:**
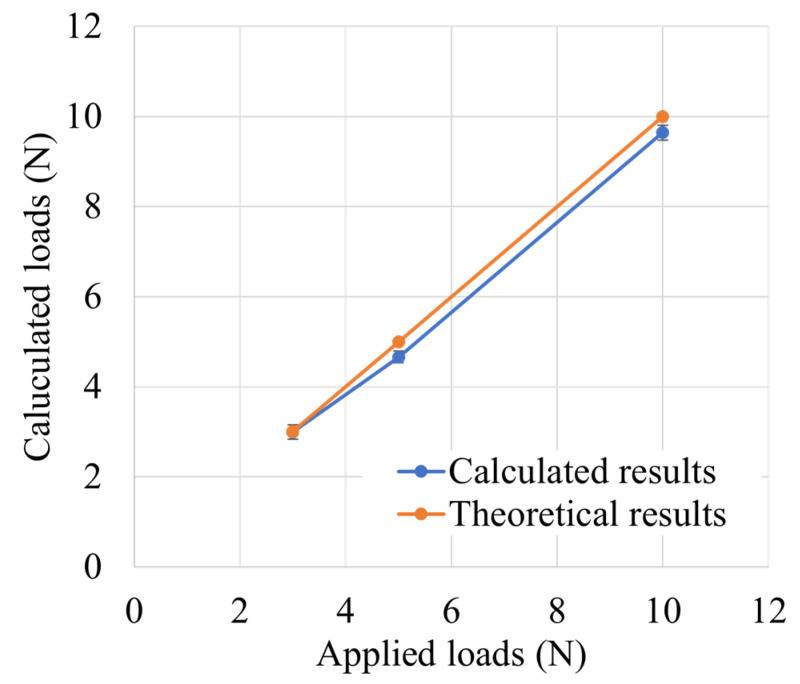
Relationship between applied and calculated load.

**Figure 8 biomimetics-09-00246-f008:**
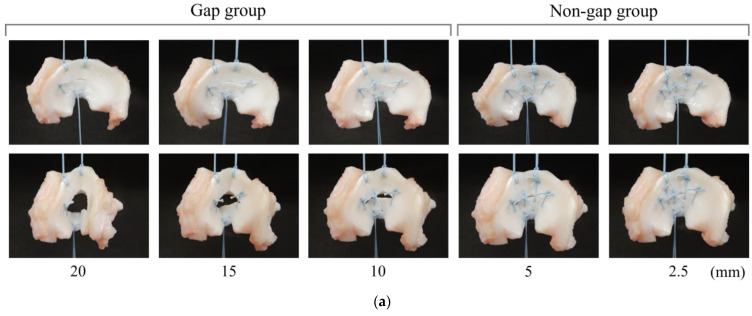
(**a**) Macroscopic images of each suture interval in biomechanical test. The upper row is an image before the test; the lower row is an image after the application of a 10 N load. (**b**) The results of a suture force test using a probing sensor after suturing a porcine meniscus (*n* = 5). (**c**) Displacement results after tensile test using the universal testing machine (*n* = 5). (**d**) The results show (**b**,**c**) side by side at each suture interval. Data are presented as average value ± SD. ✶ indicates a statistically significant difference (*p* < 0.05).

**Figure 9 biomimetics-09-00246-f009:**
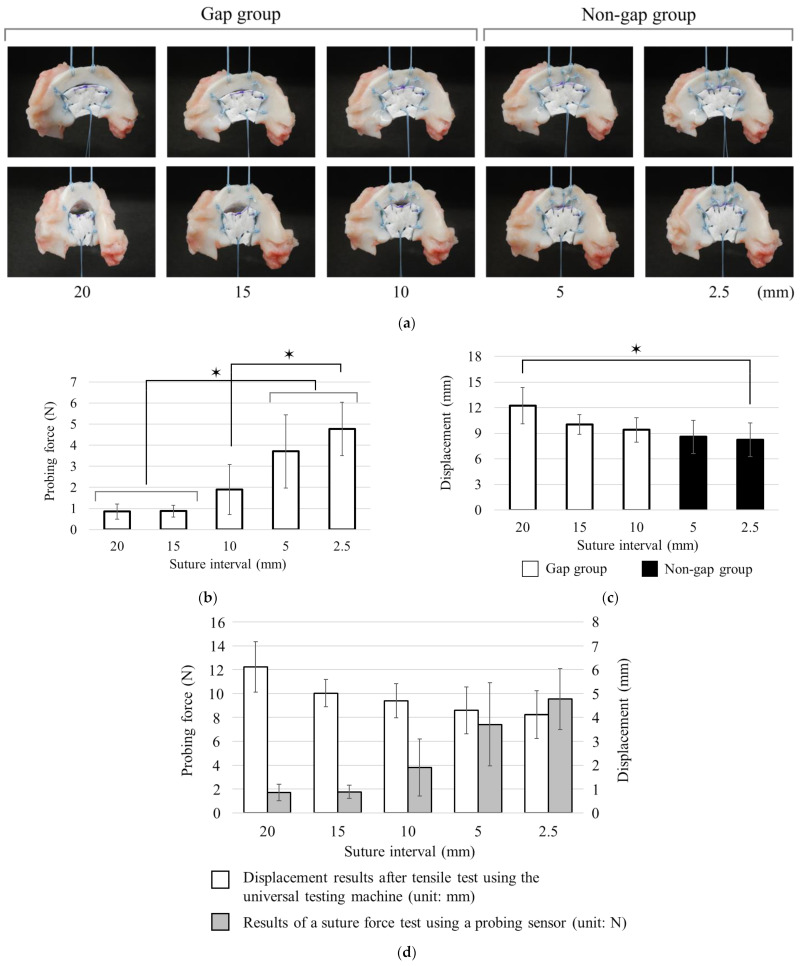
(**a**) Macroscopic images of each suture fixation interval after meniscal scaffold transplantation. The upper row is an image before the test; the lower row is an image after application of a 10 N load. (**b**) Results of a suture force test using a probing sensor after meniscal scaffold fixation (*n* = 5). (**c**) Displacement results after tensile test using the universal testing machine (*n* = 5). (**d**) The results showing (**b**,**c**) side by side at each suture interval. Data are presented as average value ± SD. ✶ indicates a statistically significant difference (*p* < 0.05).

## Data Availability

Data are contained within the article.

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
