# Peer review of "Usefulness of Probing Sensor Device for Evaluating Meniscal Suture and Scaffold Implantation"

_biomimetics, 2024, doi:10.3390/biomimetics9040246_

Round 1

Reviewer 1 Report

Comments and Suggestions for Authors

Sezaki et al. aimed to experimentally demonstrate the utility of a detection sensor device for evaluating the efficacy of half-plate suturing and stent implantation. However, several issues with the experimental design detract from the study's clarity and effectiveness:

  1. 1. The article relies on the universal testing machine as the standard reference. Beforehand, the authors should establish or elucidate the advantages of the detection sensor method over the universal testing machine to justify its use.

  2. 2. The description of suture strength and suture gap is convoluted. In Figure 8, the distinction between points b and c should be clarified, ideally by presenting them horizontally for direct comparison. Moreover, while significant differences are noted, it remains unclear whether these differences represent the detection sensor's enhanced sensitivity compared to the universal testing machine in certain areas.

  3. 3. The discussion section lacks depth. While it correctly underscores the significance of suture strength and suture gap, there's a dearth of analysis on the specific advantages offered by the detection sensor device in these assessments.

  4. 4. The conclusion diverges from the article's title. While offering practical clinical guidance is commendable, the conclusion should primarily focus on summarizing and elucidating the efficacy of the detection sensor device, aligning closely with the study's objectives.

In essence, numerous sections within the main text fail to align with the article's purpose, resulting in a disjointed logical flow. A thorough revision is warranted to address these issues effectively.

Author Response

Comments 1: The article relies on the universal testing machine as the standard reference. Beforehand, the authors should establish or elucidate the advantages of the detection sensor method over the universal testing machine to justify its use.

Response 1: Thank you for pointing this out. We agree with you. In this study, a 100N load cell installed in a universal testing machine was used. The load cell was calibrated annually to ensure that the values detected were accurate. Therefore, the load applied to the probing sensor was considered accurate. We have included this information on page 3, lines 78-80: " A universal testing machine (Autograph AGS-X, Shimadzu Corporation, Kyoto, Japan) with a load cell of 100 N was used for evaluating the reliability of the probing sensor.” and on page 3, lines 81-82: " The load cells used in this test were calibrated annually. Therefore, the values calculated by the universal testing machine were accurate.”

Comments 2: The description of suture strength and suture gap is convoluted. In Figure 8, the distinction between points b and c should be clarified, ideally by presenting them horizontally for direct comparison. Moreover, while significant differences are noted, it remains unclear whether these differences represent the detection sensor's enhanced sensitivity compared to the universal testing machine in certain areas.

Response 2: We thank you for pointing this out. In response to this comment, we created graphs in Figure 8d and Figure 9d that display the pull-out strength measured by the probing sensor and the gap distance measured by the universal testing machine side by side.

Comments 3: The discussion section lacks depth. While it correctly underscores the significance of suture strength and suture gap, there's a dearth of analysis on the specific advantages offered by the detection sensor device in these assessments.

Response 3: We appreciate you for this valid feedback. As rightly pointed out, due to the lack of analysis of the advantages offered by the device, and we have revised the statement on page 10, lines 280-284: "It was thought that this device could turn qualitative arthroscopy, which had been termed as "surgeon's sense", into a quantitative evaluation. Based on this data, if clinically appropriate pull-out strength is evaluated in further research using this device, the re-operation rate after meniscus suturing may be reduced." and page 11, lines 296-298: "This information indicated that similar to meniscal repair, the use of this device in clinical practice may provide superior fixation for meniscal scaffold grafts and reduce their failure rate."

Comments 4: The conclusion diverges from the article's title. While offering practical clinical guidance is commendable, the conclusion should primarily focus on summarizing and elucidating the efficacy of the detection sensor device, aligning closely with the study's objectives.

Response 4: We thank you for this comment. In response to this, we have changed the conclusion to match the purpose and focus on the efficacy of the detection sensor device. Furthermore, we have changed the purpose to a clearer statement. We have revised the statement on page 2, lines 58-62: “Therefore, this study aimed to evaluate whether the probing sensor could be a reliable device, as a basic study, for assessing its clinical application and usefulness. We also aimed to determine the appropriate pull-out strength by this device during meniscal suture repair and scaffold implantation using porcine meniscus.” and page 11, lines 333-336: "This information demonstrated adequate meniscal suture fixation force through in vitro evaluation. With further investigation, this device may be able to contribute to reducing the failure rate of meniscal suture repair and meniscal scaffolding in clinical practice."

Reviewer 2 Report

Comments and Suggestions for Authors

It is an interesting manuscript. However, needs some modifications before can be accepted.

1. The research gap is unclear. Authors should state the current devices that can evaluate the meniscal and scaffold implantation. Put it in the introduction section.

2. Jig - please include a picture of the jig. From figure 2, it is not clear the share of the jig.

3. Need to include the load rate and load cell used for the tensile and compression test.

4. Any ISO standard that you refer to, for the tensile and compression test?

5. Ethical approval for using the porcine? Need to include. Please provide.

6. Line 109 - I believe you should consider ISO 10334:1994 for your testing procedures. 

7. Discussion - authors should compare their results with other from literature.

Comments on the Quality of English Language

minor grammatical errors found

Author Response

Comments 1: The research gap is unclear. Authors should state the current devices that can evaluate the meniscal and scaffold implantation. Put it in the introduction section.

Response 1: Thank you for your comment. In clinical use, we believe that there is a gap in the lack of devices that can be used to identify the residual force of appropriate suture tension evaluated during meniscal repair and meniscal scaffold fixation.

Comments 2: Jig - please include a picture of the jig. From figure 2, it is not clear the share of the jig.

Response 2: We appreciate this feedback. In response to this comment, we have re-taken the test image and replaced Figure 2.

Comments 3: Need to include the load rate and load cell used for the tensile and compression test.

Response 3: Thank you for pointing this out. We agree with this comment. Therefore, we have included this information on that a 100 N load cell was used in this test, and a load rate of 1.0 mm/s was applied in the reliability of the probing sensor test. We have included this information in different sections of the manuscript: page 3, lines 78-80, page3, lines 81-82, page 3, line 88, page 4, lines 123.

Comments 4: Any ISO standard that you refer to, for the tensile and compression test?

Response 4: Due to the nature of this test, there were no ISO standards referenced.

Comments 5: Ethical approval for using the porcine? Need to include. Please provide.

Response 5: Regarding your request on the research paper we provided, there was no specific approval made by the Ethical Committee. The porcine knee used for the experiment was purchased from the slaughterhouse (Kyoto Kyodo Kanri Co., Ltd., Manuscript Line: 96-97), and was originally intended for disposal. In this circumstance of purchase, an approval of the Ethical Committee is not required, hence the absence of information on the matter.

In addition, our previous study used similar porcine knee without requiring Ethical Committee approval (Sezaki et al. J Biomed Mater Res B Appl Biomater. 2023. doi: 10.1002/jbm.b.35199. Epub 2022 Nov 14.).

We have included this information on page 3, lines 99-102: " The porcine knee used for the experiment was purchased from the slaughterhouse (Kyoto Kyodo Kanri Co., Ltd., Kyoto, Japan), and was originally intended for disposal. In this circumstance of purchase, an approval of the Ethical Committee was not re-quired, hence the absence of information on the matter."

Comments 6: Line 109 - I believe you should consider ISO 10334:1994 for your testing procedures.

Response 6: In this study, we have evaluated the stability of meniscal tear repair and meniscal scaffold fixation in a clinical setting, without measuring the strength of the suture itself. Therefore, ISO 10334:1994 was not considered as it did not measure the strength of the suture itself.

Comments 7: Discussion - authors should compare their results with other from literature.

Response 7: Thank you for your comment. It was a perspective that was missing in the discussion. We have added a comparison with results from other literature. We have included this information on page 11, lines 306-319: "In other areas of orthopedics, as mentioned in the background, this device is be-ing applied to quantitatively evaluate the hip [23] and knee joints [24]. Regarding applied use to the hip joint, no difference in the stiffness of the hip labrum between in vivo and cadaveric tissue has been reported; the calculated value using the probing device was 4.2–8.3 N [23]. On the other hand, in a study evaluating anterior cruciate ligament (ACL) reconstruction, a moderate correlation between the amount of strain caused by the resistance force of the ACL using a probing sensor and the tensile ten-sion of the ligament was shown; the value calculated using the probing device was 0.67–1.54 N [24]. Although a simple comparison is not possible, the appropriate pull-out strength for meniscal repair or meniscal scaffold fixation obtained in our study may be a reasonable value considering that the meniscus was a cartilage-like tissue. Additionally, Tuijthof et al. reported in a probing force test using a meniscal model that forces exceeding 8.5 N should be avoided when probing meniscal tissue [35]. There may also be room to consider the upper limit of probing force.”

Round 2

Reviewer 1 Report

Comments and Suggestions for Authors

Accept in present form

Author Response

Thank you very much for taking the time to review this manuscript.

Reviewer 2 Report

Comments and Suggestions for Authors

Last comment from me.

The research gap should be highlighted clearly in the introduction section. Not only response to my comments. Please include it in your introduction section.

All other comments have been carefully addressed by the authors.

Author Response

Comments: The research gap should be highlighted clearly in the introduction section. Not only response to my comments. Please include it in your introduction section.

All other comments have been carefully addressed by the authors.

Response: Thank you for pointing this out. As per the comment, the description of the research gap was insufficient, and we have revised the statement on page 1, lines 44-48: "Although regulating suture tension for meniscal repair or implantation is essential to improve the success rate of meniscal sutures, there is no device that can quantitatively assess the suture tension in clinical practice. Therefore, suture tension for meniscal repair or implantation only relies on “surgeon's sense” and controlling suture tension remains a challenge."
